# Gli1 identifies osteogenic progenitors for bone formation and fracture repair

Yu Shi[1], Guangxu He[1,2], Wen-Chih Lee[1], Jennifer A. McKenzie[1], Matthew J. Silva[1] & Fanxin Long[1,3]

Bone formation in mammals requires continuous production of osteoblasts throughout life. A common molecular marker for all osteogenic mesenchymal progenitors has not been identified. Here, by lineage-tracing experiments in fetal or postnatal mice, we discover that Gli1[+] cells progressively produce osteoblasts in all skeletal sites. Most notably, in postnatal growing mice, the Gli1[+] cells residing immediately beneath the growth plate, termed here "metaphyseal mesenchymal progenitors" (MMPs), are essential for cancellous bone formation. Besides osteoblasts, MMPs also give rise to bone marrow adipocytes and stromal cells in vivo. RNA-seq reveals that MMPs express a number of marker genes previously assigned to mesenchymal stem/progenitor cells, including CD146/Mcam, CD44, CD106/Vcam1, Pdgfra, and Lepr. Genetic disruption of Hh signaling impairs proliferation and osteoblast differentiation of MMPs. Removal of β-catenin causes MMPs to favor adipogenesis, resulting in osteopenia coupled with increased marrow adiposity. Finally, postnatal Gli1[+] cells contribute to both chondrocytes and osteoblasts during bone fracture healing. Thus Gli1 marks mesenchymal progenitors responsible for both normal bone formation and fracture repair.

[1] Department of Orthopedic Surgery, Washington University School of Medicine, St. Louis, MO 63110, USA. [2] Department of Orthopedic Surgery, The Second Xiangya Hospital, Central South University, Hunan 410011, China. [3] Departments of Medicine and Developmental Biology, Washington University School of Medicine, St. Louis, MO 63110, USA. Correspondence and requests for materials should be addressed to F.L. (email: flong@wustl.edu)

Osteoblastogenesis in humans and mice occurs throughout life during not only bone modeling that begins in the embryo and continues well after birth but also bone remodeling that takes place in the mature skeleton. However, the molecular identity of the osteogenic mesenchymal progenitors is not well understood. Extensive work has been devoted to the postnatal bone marrow stromal cells (BMSCs) as they were historically shown to contain cells, now commonly referred to as mesenchymal stem cells (MSCs), that can proliferate in vitro to form fibroblast colonies, which in turn can produce bone and cartilage in diffusion chambers or generate bone organs complete with a hematopoietic marrow when grafted in vivo[1–5]. Work in the past two decades with cell sorting techniques has revealed that cell surface markers for MSCs are likely complex and variable especially following expansion in vitro[6, 7]. Nonetheless, Pdgfra and Sca-1 have been useful for prospective enrichment of MSCs in the mouse[8]. Recently, an AlphaV+CD200+ cell population isolated from the bones of newborn mice was shown to posses the capacity to form bone, cartilage, and stromal cells when transplanted into the kidney capsules of immunodeficient mice, but how this population relates to the bone marrow MSCs is not clear[9].

Recent experiments with lineage-tracing techniques have provided important insights about skeletal progenitor cells in vivo. Perinatal Sox9+, Col2+, Acan+, or Osx+ cells have all been shown to produce osteoblasts and BMSCs in both growing and adult (4 months of age) mice[10, 11]. However, it is not clear how those perinatal progenitors relate to the Lepr+ stromal cells that

generate osteoblasts mainly in adult mice[12, 13]. Distinct from the Lepr+ stromal cells that are present throughout the marrow cavity, a Gremlin-positive (Grem1+) progenitor population was found to reside underneath the growth plate but appeared to have limited contribution to the adult bone[14]. In addition, αSMA+ cells in juvenile mice (4–5 weeks of age) have been shown to produce osteoblasts, but their long-term contribution to bone was not determined[15]. Overall, the prior work has highlighted the heterogeneity and complexity of skeletal progenitors in postnatal mice.

Indian Hedgehog (Ihh) signaling critically regulates osteoblast differentiation during endochondral bone development in the embryo. Ihh, one of the three Hedgehog (Hh) proteins in mammals, signals via the seven-pass transmembrane protein Smoothened (Smo) to modulate gene expression by either activating or derepressing the Gli1–3 transcription factors[16]. Interestingly, Gli1 is not only a transcription activator but also a target gene of Gli proteins, thus amplifying the transcriptional response to Hh signaling. Ihh regulates osteoblast differentiation both indirectly by controlling cartilage development and directly through signaling to the perichondrial progenitors[17, 18]. Moreover, the direct osteogenic function of Ihh involves both activation of Gli2 and derepression of Gli3 and is further augmented by Gli1[19, 20]. Hh signaling has also been implicated in bone formation in postnatal mice, as deletion of Ihh from the growth plate chondrocytes in newborn mice caused continuous loss of trabecular bone at an older age[21]. Conversely, forced activation of Hh signaling caused by either global Ptc1 haploinsufficiency or Ptc1

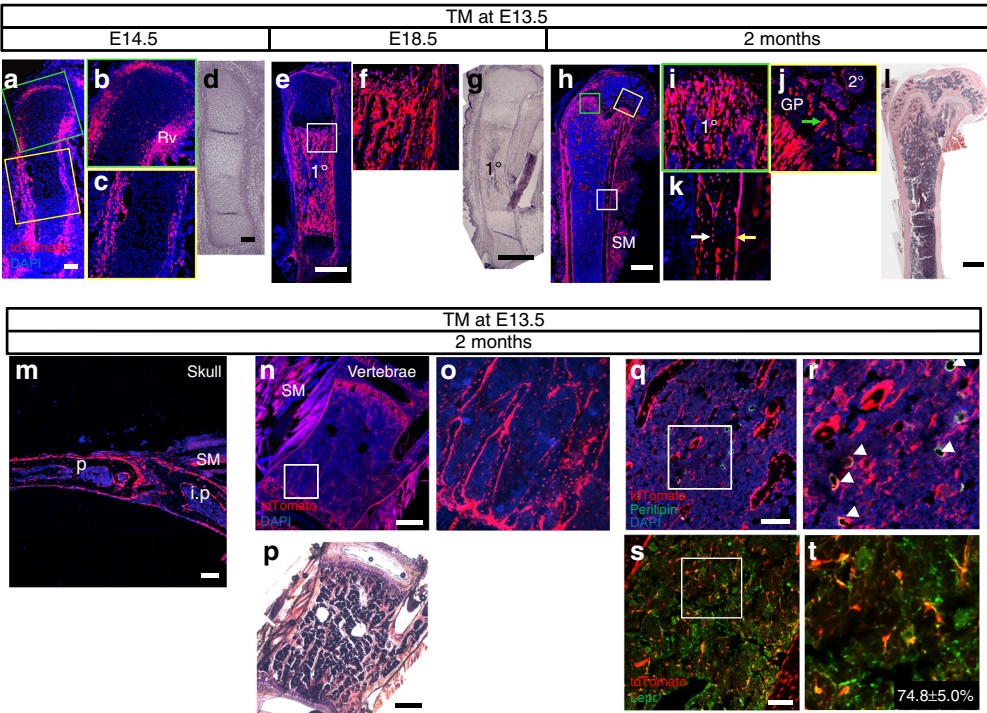

**Fig. 1** Embryonic Gli1+ cells give rise to multiple cell types in postnatal mice. Gli1-CreER^T2; Ai9 mice were administered tamoxifen (TM) at E13.5 and harvested at E14.5 **a–d**, E18.5 **e–g**, or 2 months of age **h–l**. Red: tdTomato+ cells; blue: nuclear staining by DAPI. Scale bars: 100 μm **a**, **d**, **m**, **q**, **s** or 500 μm **e**, **g**, **h**, **l**, **n**, **p**. **a–c**, **e–f**, **h**, **k** Representative confocal images from frozen sections of the femur. Boxed areas are shown at a higher magnification in corresponding panels to the right **b**, **c**, **i–k**. Rv Ranvier's groove, SM skeletal muscle, GP growth plate. 1°: primary ossification center; 2°: secondary ossification center. Green arrow: chondrocytes in growth plate; white arrow: osteocyte; yellow arrow: osteoblast on periosteal surface. **d**, **g**, **l** H&E staining of sister sections corresponding to **a**, **e**, and **h**, respectively. **m–o** Representative confocal images from frozen sections through the lambdoid suture **m** or lumbar vertebrae **n**, **o**. Boxed area in **n** is shown at a high magnification in **o**. P parietal bone, Ip interparietal bone. **p** H&E staining of an adjacent section to **n**. **q–t** Immunofluorescence staining of Perilipin **q**, **r** or Lepr **s**, **t** on frozen sections of distal metaphyseal **q**, **r** or diaphyseal **s**, **t** bone marrow region of the femur. Boxed areas in **q** and **s** are shown at a high magnification in **r** and **t**, respectively. Arrowheads in **r** denote tdTomato+ adipocytes. Numbers in **t** indicate percentage (mean ± SD, n = 3) of tdTomato+Lepr+ over total tdTomato+ cells on two sections per mouse and three mice for each genotype

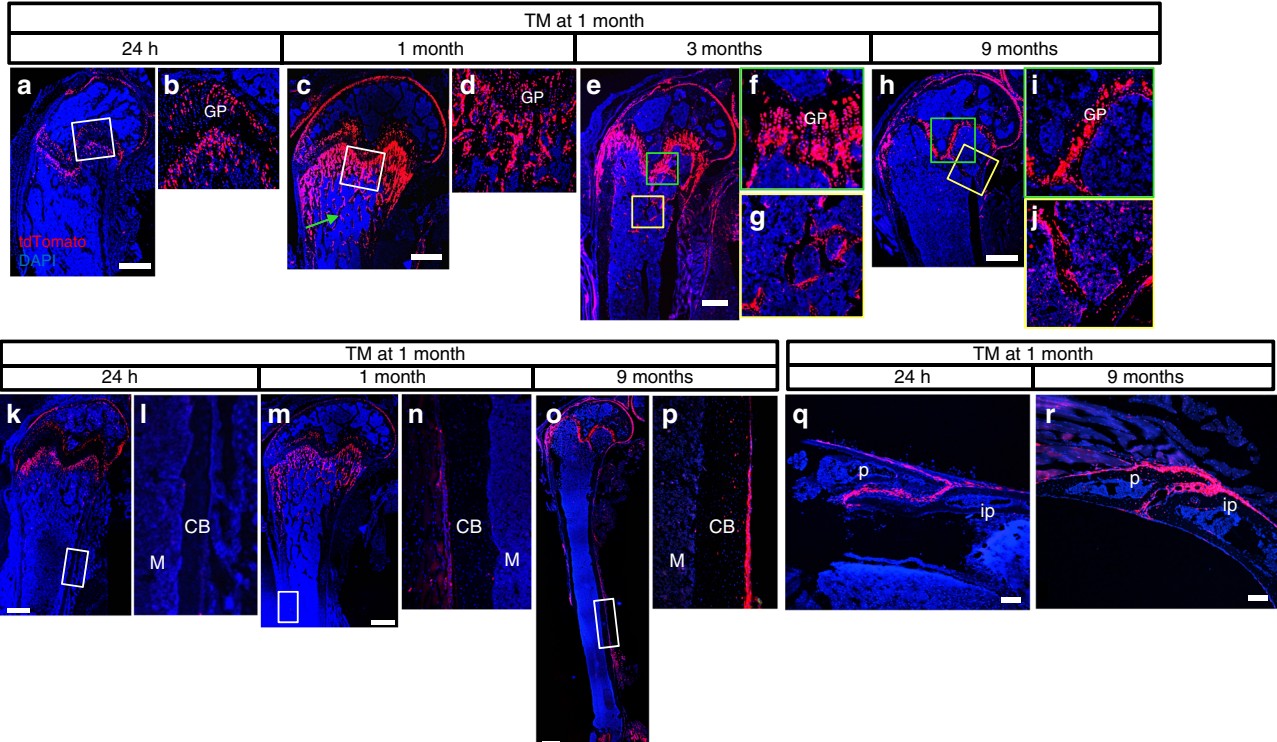

**Fig. 2** Postnatal Gli1[+] cells contribute to osteoblasts in both endochondral and intramembranous bones. Gli1-CreER[T2]; Ai9 mice were treated with TM at 1 month of age and analyzed at 24 h **a**, **b**, **k**, **l**, **q**; 1 month **c**, **d**, **m**, **n**; 3 months **e**–**g**; or 9 months **h**–**j**, **o**, **p**, **r** after the last TM dosing. Representative images from three mice for each time point are presented. Red: tdTomato[+] cells; blue: nuclear staining by DAPI. Scale bars: 500 μm **a**–**o** or 100 μm **q**, **r**.
**a**–**p** Representative confocal images of frozen sections of the femur. Boxed areas are shown at a high magnification in corresponding panels to the right. GP growth plate. Green arrow: trabecular bone. CB cortical bone, M marrow. **q**, **r** Representative confocal image showing sections through the lambdoid suture of the skull. P parietal bone, Ip interparietal bone

deletion in mature osteoblasts enhanced bone formation in postnatal mice[22, 23]. These studies, however, do not address the physiological function of Hh signaling in the maintenance or differentiation of osteoblast progenitors in postnatal mice. By utilizing the Gli1-CreER[T2] mouse expressing a tamoxifen (TM)-inducible form of Cre (CreER[T2]) from the endogenous Gli1 locus, recent studies have indicated that Gli1 marks "MSCs" in the craniofacial bones, the incisor as well as several internal organs of adult mice[24–27]. However, the potential relationship between Gli1 and postnatal skeletal progenitors in the long bones has not been investigated.

Here we demonstrate that Gli1[+] cells progressively produce essentially all osteoblasts in the murine skeleton. Whereas osteoblasts in the secondary ossification center and the bone shaft are mostly descendants of embryonic Gli1[+] cells, the cancellous bone osteoblasts in the primary ossification center of postnatal mice are produced from Gli1[+] "metaphyseal mesenchymal progenitors" (MMPs) residing immediately below the growth plate. RNA profiling of the MMPs by RNA-seq reveals a molecular signature enriched with multiple cell-surface markers associated with MSCs, including CD146/Mcam, CD44, and CD106/Vcam1. In addition, postnatal Gli1[+] cells contribute to chondrocytes and osteoblasts during bone fracture healing. Thus Gli1-lineage cells encompass osteogenic mesenchymal progenitors responsible for not only normal bone formation but also fracture repair.

## Results
**Gli1-lineage cells are principle precursors for osteoblasts**. To investigate the contribution of Hh-responsive cells to the skeleton, we traced the fate of Gli1-expressing cells in mice with the genotype of Gli1-CreER[T2]; Ai9 by administering TM at the

selected times before harvest at different ages. Previous studies have indicated that nuclear Cre activity from the Gli1-CreER[T2] allele is limited to cells actively responding to Hh at the time of TM administration or approximately 24 h after[27]. We term those cells Gli1[+] cells. In our system, both Gli1[+] cells and their descendants (Gli1-lineage cells) are permanently marked by expression of the red fluorescent protein tdTomato. In the first set of experiments, we administered TM at E13.5 before the appearance of mature osteoblasts in the endochondral skeleton and harvested the mice at E14.5, E18.5, or postnatal 2 months of age. At E14.5, Gli1[+] cells were found predominantly in the perichondrium encasing the cartilage anlage, with relatively few in the cartilage proper (Fig. 1a, d). The perichondrial Gli1[+] cells were present in multiple layers covering the joint surface, the Ranvier's groove, and the shaft where osteoblasts first form around E14.5 (Fig. 1b, c). The abundance of Gli1[+] cells in the perichondrium corresponded well with Gli1 mRNA expression at E14.5 as previously detected by in situ hybridization, but the number of Gli1[+] chondrocytes observed here was lower than expected, perhaps due to poor availability of TM to the cartilage proper at this stage[18]. When the mice were analyzed at E18.5, tdTomato[+] cells were most prominent in the primary ossification center occupying the mid-section of the long bone (Fig. 1e–g). At 2 months after birth, tdTomato[+] cells were abundant throughout the cortical bone, as well as the cancellous bone in both primary and secondary ossification centers of the long bone (Fig. 1h, l, i–k). High-magnification images indicated that the positive cells included both osteoblasts on the bone surface and osteocytes embedded in the bone matrix, as well as some growth plate chondrocytes (Fig. 1j, k). Similarly, most osteoblasts and osteocytes in the skull and vertebrae expressed tdTomato (Fig. 1m–p).

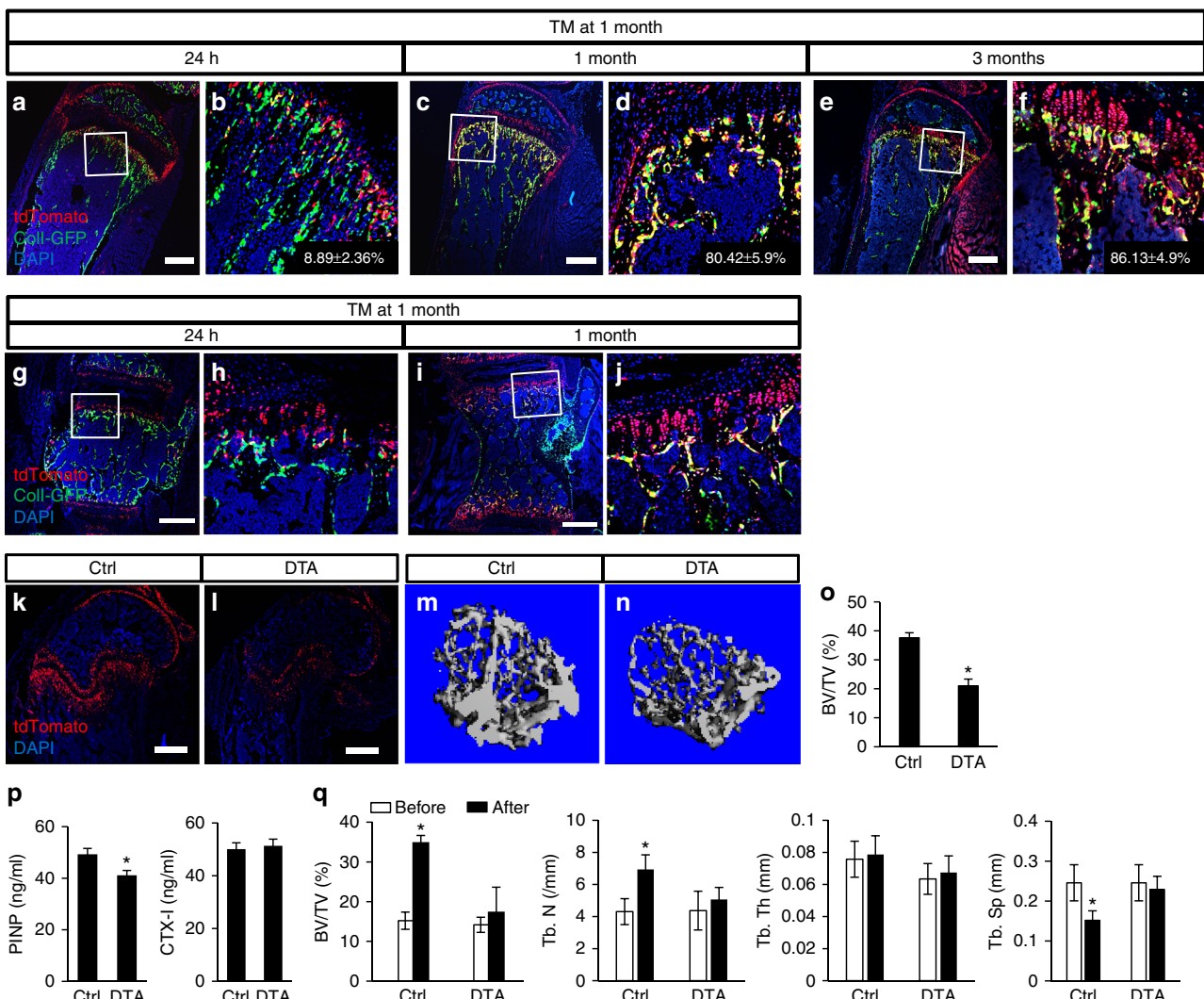

**Fig. 3** Gli1[+] cells are critical for cancellous bone formation in postnatal mice. **a–j** Postnatal Gli1[+] cells produce osteoblasts. Gli1-CreER[T2]; Ai9; ColI-GFP mice were treated with TM at 1 month of age and analyzed at 24 h **a**, **b**, **g**, **h**; 1 month **c**, **d**, **i**, **j**; or 3 months **e**, **f** after the last TM dosing. Representative confocal images are shown for tibia **a–f** and lumbar vertebrae **g–j**. Boxed areas are shown at a higher magnification in corresponding panels to the right. Percentage (mean ± SD, n = 3) of tdTomato[+]GFP[+] over total GFP[+] cells **a–f** was calculated from cancellous bone region extending 300 μm from growth plate on two sections each of three mice. Scale bar: 500 μm. **k–q** Postnatal Gli1[+] cells are necessary for cancellous bone formation. DTA (Gli1-CreER[T2]; Ai9; Rosa-DTA) or Ctrl (Gli1-CreER[T2]; Ai9) littermates were treated with TM once daily for 3 consecutive days starting at 1 month of age and harvested at 3 weeks after last dosing. **k**, **l** Loss of tdTomato[+] cells in the distal femur of DTA (**l**) compared to Ctrl mice (**k**) at the time of harvest. Scale bar: 500 μm. **m**, **n** Representative μCT images from cancellous bone of distal femur. **o** Quantification of cancelluous bone mass (BV/TV) by μCT in distal femur. **p** Serum levels of P1NP (left) and CTX-I (right) at the time of harvest. **q** In vivo microCT analyses of cancellous bone parameters in distal femur of mice immediately before TM (white bar, "Before") or immediately before harvest (black bar, "After"). BV/TV bone volume over tissue volume, Tb.N trabeculae number, Tb.Th trabeculae thickness, Tb.Sp trabeculae spacing. n = 3, *p < 0.05, paired Student's t-tests

At all the skeletal sites examined, some muscle fibers were also positive (Fig. 1h–p). Within the bone marrow of long bones, tdTomato marked many adipocytes as identified by perilipin immunostaining and also contributed to the stromal component (Fig. 1q, r). Interestingly, 70–80% of the tdTomato[+] BMSCs expressed leptin receptor (Lepr) (Fig. 1s, t). Thus embryonic Gli1[+] cells give rise to multiple cell types associated with the skeleton and are a major source of osteoblasts in both fetal and postnatal life of the mouse.

We next examined whether postnatal Gli1[+] cells continued to be a source of osteoblasts. To this end, we administered three daily doses of TM to the Gli1-Cre[ERT2]; Ai9 mice starting at 1 month of age and tracked the tdTomato[+] cells after 1, 3, or 9 months of chasing. Analyses at 24 h after the last dosing identified Gli1[+] cells in four distinct domains: the articular

cartilage, the upper layers of the growth plate, the perichondrium flanking the growth plate, as well as the chondro-osseous junction immediately beneath the growth plate (Fig. 2a, b). No tdTomato was detected in either cortical or cancellous bone or within the marrow, indicating no detectable Hh signaling in mature osteoblasts or osteocytes or the bone marrow cells. Notably, after 1 month of chasing, not only did the tdTomato[+] domain below the growth plate expand considerably but the signal also extended to the surface of numerous bone trabeculae within the primary ossification center, likely indicating the production of tdTomato[+] osteoblasts (Fig. 2c). The tdTomato[+] chondrocytes also expanded but did not yet reach the full thickness of the growth plate at this time (Fig. 2d). After 3 months of chasing, tdTomato was detected throughout the height of the growth plate, whereas the positive domain under the growth plate was reduced from that of the

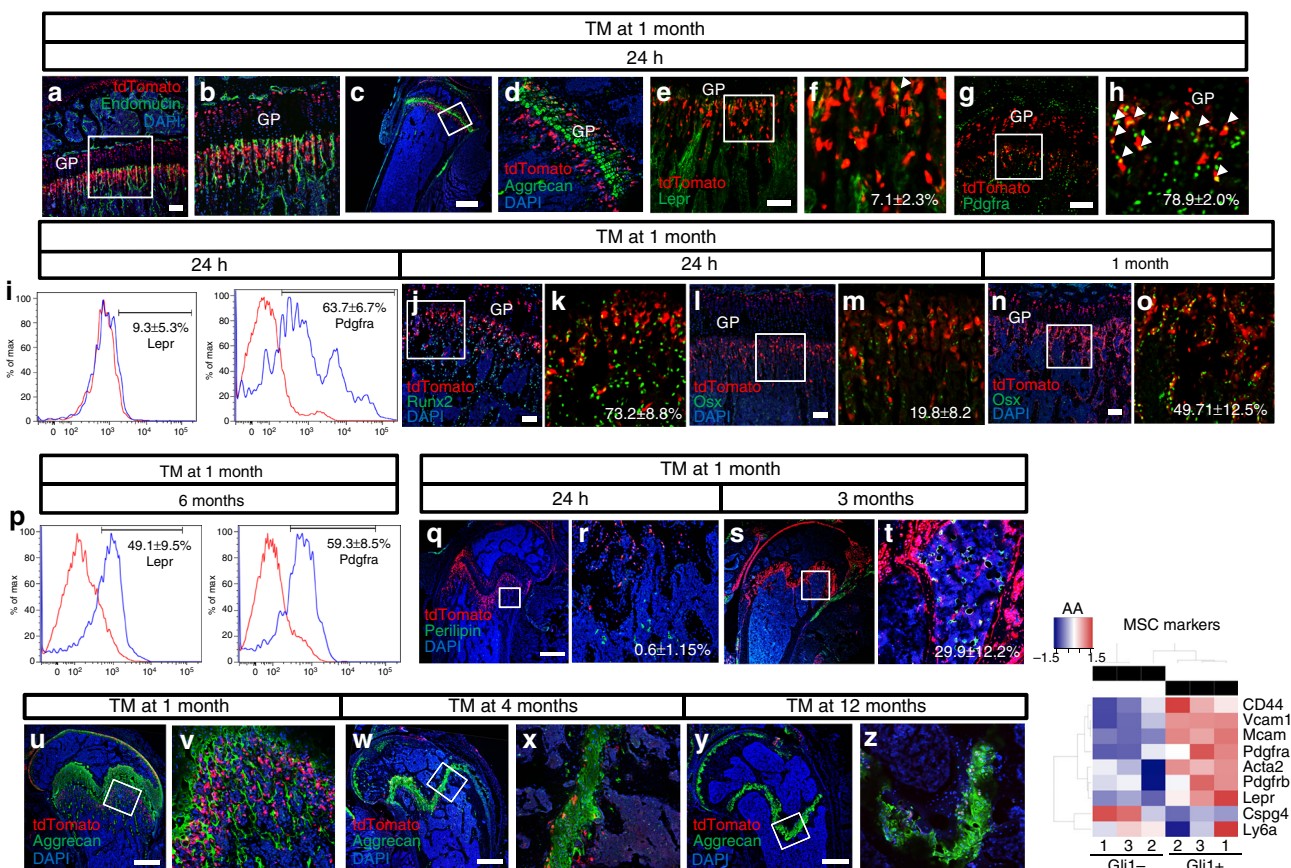

**Fig. 4** Gli1 marks MMPs in postnatal growing mice. **a–f** Confocal images of immunofluorescence staining for Endomucin **a**, **b**, Aggrecan **c**, **d**, or Lepr **e**, **f** on frozen sections of the proximal tibia. Boxed areas shown in magnified images to the right. GP growth plate. Arrowhead in **f** denotes a typical Lepr⁺tdTomato⁺ cell. Percentage (mean ± SD, n = 3) of double positive cells over total tdTomato⁺ cells was calculated from cancellous bone region extending 300 μm from growth plate in two sections each of three mice. Same below. Scale bars: 100 μm (**a**, **e**) or 500 μm **c**. **g**, **h** Confocal images of direct fluorescence on frozen sections of proximal tibia. Boxed area shown at a higher magnification to the right. Arrowheads in **h** denote Pdgfra⁺tdTomato⁺ cells. Scale bar: 100 μm **g**. **i** FACS analyses of Lepr (left) or Pdgfra (right) expression among CD31⁻CD45⁻Ter119⁻tdTomato⁺ cells from the cancellous bone region beneath the growth plate of either distal femur or proximal tibia. Red and blue lines represent control IgG and specific antibody, respectively. **j–o** Confocal images of Runx2 **j**, **k** or Osx **l–o** immunofluorescence staining on frozen sections of the proximal tibia. Boxed areas are shown in magnified images to the right **k**, **m**, **o**. DAPI signal was omitted in **k**, **m**, **o** for better visualization. Scale bar: 100 μm. **p** FACS analyses of Lepr (left) or Pdgfra (right) expression among CD31⁻CD45⁻Ter119⁻tdTomato⁺ from the bone marrow. Red and blue lines represent the control and specific antibody, respectively. Quantification is presented as mean ± SD, n = 3. **q–t** Confocal images of immunofluorescence staining for perilipin on frozen sections of distal femur. Boxed areas are shown at higher magnification to the right. Percentage (mean ± SD, n = 3) of perilipin⁺tdTomato⁺ over perilipin⁺ adipocytes was acquired from cancellous bone region extending 400 μm from the growth plate in two sections each of three mice. Scale bar: 500 μm. **u–z** Confocal images showing Gli1⁺ cells (red) and immunofluorescence signal of Aggrecan (green) in distal femur of Gli1-CreER^T2;Ai9 mice treated with TM at different ages and harvested after 24 h. Scale bar: 500 μm. Boxed areas are shown at higher magnification to the right **v**, **x**, **z**. (AA) Differential expression of MSC marker genes in Gli1⁺ MMPs vs. Gli1⁻ mesenchymal cells. Heatmap generated from RNA-seq data of three paired biological replicates. Scale derived from voom-transformed data indicating relative increase (red) or decrease (blue) in MMPs

earlier time point (Fig. 2e, f). Importantly however, tdTomato continued to decorate the surface of bone trabeculae in the primary ossification center (Fig. 2g). At 9 months after TM dosing, the growth plate remained tdTomato⁺ but was reduced in height (Fig. 2h, i). Moreover, the tdTomato⁺ domain under the growth plate all but disappeared, concurrent with a marked decrease of bone trabeculae in the primary ossification center at this age (Fig. 2i). Nonetheless, most osteoblasts and osteocytes of the remaining bone trabeculae expressed tdTomato (Fig. 2j). Distinct from the earlier time points, tdTomato⁺ cells became notable in the bone marrow at this time. In contrast, no tdTomato was detected in either the bony tissue or marrow of the secondary ossification center at any time point following TM at 1 month. Throughout 9 months of chasing, the tdTomato⁺ articular chondrocytes consistently occupied the 3–4 cell layers closest to the joint surface. In addition, although no tdTomato

was detected in the cortical bone at 24 h after TM, positive cells appeared in certain areas of the periosteum after 1 month of chasing (Fig. 2k–n). After 9 months, tdTomato spread to more areas of the periosteal surface (Fig. 2o, p). More importantly, some osteocytes within the cortical bone expressed tdTomato at this age (Fig. 2p). Finally, in the skull, tdTomato⁺ cells were initially concentrated in the interior of the lambdoid suture or the posterior frontal suture at 24 h after TM (Fig. 2q, Supplementary Figure 1). After 9 months of chasing, tdTomato clearly marked both the bone surface and the osteocytes near the lambdoid suture (Fig. 2r). Overall, the Gli1⁺ cells in postnatal growing mice continue to produce osteoblasts in both endochondral and intramembranous bones.

We next chose to focus our study on the Gli1⁺ cells below the growth plate. To corroborate that those cells give rise to osteoblasts in the primary ossification center, we fate-mapped

**Table 1 Relative expression of MSC cell surface markers in Gli1+ MMPs over Gli1− mesenchymal cells isolated from the chondro-osseous junction of one-month-old mice**

| gene_name | logFC | CI.L | CI.R | AveExpr | t | adj.P.Val | LinearFC |
|-----------|-------|------|------|---------|---|-----------|----------|
| Vcam1 | 1.43E+00 | 1.238381579 | 1.62E+00 | 6.90E+00 | 1.73E+01 | 2.44E−06 | 2.689604 |
| Mcam | 1.15E+00 | 0.623453805 | 1.67E+00 | 1.06E+00 | 5.02E+00 | 2.98E−03 | 2.214369 |
| Cd44 | 6.68E−01 | 0.570826489 | 7.64E−01 | 8.51E+00 | 1.58E+01 | 3.97E−06 | 1.588379 |
| Pdgfra | 1.06E+00 | 0.535143995 | 1.58E+00 | 1.44E+00 | 4.64E+00 | 4.53E−03 | 2.082435 |
| Pdgfrb | 4.41E−01 | 0.090816963 | 7.91E−01 | 1.88E+00 | 2.89E+00 | 3.89E−02 | 1.357405 |
| Lepr | 7.36E−01 | 0.252759698 | 1.22E+00 | 1.21E+00 | 3.49E+00 | 1.77E−02 | 1.665826 |
| Acta2 | 3.32E−01 | 0.094708473 | 5.68E−01 | 3.43E+00 | 3.21E+00 | 2.54E−02 | 1.258329 |
| Cspg4 | −1.59E+00 | −1.884728395 | −1.30E+00 | 5.21E+00 | −1.24E+01 | 1.53E−05 | −3.012349 |
| Ly6a | −3.33E−02 | −0.358501759 | 2.92E−01 | 3.02E+00 | −2.34E−01 | 8.63E−01 | −1.023329 |

RNA-seq results from three paired pools of Gli1+ vs. Gli1− mesenchymal cells isolated from the chondro-osseous junction of one-month-old mice
Adj.P.Val: adjusted *p*-value
LinearFC: linear fold change (Gli1+ over Gli1− cells)
logFC: estimate of the log2-fold-change
CI.L: left limit of confidence interval for logFC
CI.R: right limit of confidence interval for logFC
AveExpr: average log2-expression for the feature (gene or isoform) over all samples
t: moderated *t*-statistic
adj.P.Value: adjusted *p*-value using the Benjamini–Hochberg method (most stringent statistical significance). *p* < 5.00E−02 considered statistically significant
LinearFC: estimate of the linear fold-change

Gli1-linege cells in mice expressing green fluorescent protein (GFP) specifically in osteoblasts. Specifically, we generated mice with the genotype of Gli1-CreER[T2]; Ai9; ColI-GFP and applied TM at 1 month of age. Analyses at 24 h confirmed that GFP marked many osteoblasts of the cancellous bone as expected (Fig. 3a). Importantly, <10% of the tdTomato[+] cells under the growth plate co-expressed GFP, indicating that most of the Gli1[+] cells were not mature osteoblasts (Fig. 3b). However, after 1 month of chasing, >80% of the tdTomato[+] cells now expressed GFP (Fig. 3c, d). As the tdTomato[+] chondrocytes had not yet reached the chondro-osseous junction after 1-month chasing, the double-positive osteoblasts were most likely derived from the Gli1[+] cells below the growth plate. After 3 months of chasing, the percentage of double positive among tdTomato[+] cells below the growth plate remained similar (~86%), even though now entire columns of growth plate chondrocytes expressed tdTomato (Fig. 3e, f). Similarly, in vertebrae, the Gli1[+] cells beneath the cartilage endplate were initially negative for GFP but differentiated to GFP[+] osteoblasts after 1 month of chasing (Fig. 3g–j). Thus the Gli1[+] cells under the growth plate are a major source of osteoblasts in supporting cancellous bone formation in postnatal mice.

To assess the contribution of the osteogenic Gli1[+] cells to bone mass, we genetically ablated those cells by inducing expression of the cytotoxic diphtheria toxin A (DTA). Specifically, we applied TM to either Gli1-CreER[T2];Ai9;Rosa-DTA (DTA mice) or Gli1-CreER[T2];Ai9 mice (Ctrl mice) at 1 month of age and harvested the bones after 3 weeks. Sections of the distal femur from the DTA mice contained markedly fewer tdTomato[+] cells than the Ctrl mice, confirming the effectiveness of the cell ablation technique (Fig. 3k, l). Importantly, three-dimensional (3D) reconstruction of μCT images revealed that the cancellous bone was notably less in the DTA mice (Fig. 3m, n). Quantification confirmed a significant decrease in bone volume/tissue volume (BV/TV) in the DTA mice (Fig. 3o). The deficit in bone mass was likely due to defective bone formation as the serum propeptide of type I procollagen (P1NP) level was significantly lower in the DTA mice immediately before sacrifice, whereas the bone resorption marker C-telopeptide (CTX-I) was similar between the genotypes (Fig. 3p). To monitor potential changes in the cancellous bone of each mouse during the 3 weeks, we performed in vivo μCT with the distal femur immediately before TM treatment and again before the harvest. Whereas the Ctrl mice exhibited a notable increase in bone mass (BV/TV) and

trabecular number (Tb. N) and decrease in trabecular spacing (Tb. Sp), the DTA mice showed no changes in those parameters during this time (Fig. 3q). Together, the data so far support that the Gli1[+] cells below the growth plate are functionally important for supporting cancellous bone formation in postnatal mice.

**Gli1 marks postnatal "metaphyseal mesenchymal progenitors".** We then sought to characterize the osteogenic Gli1[+] cells under the growth plate in 1-month-old mice. Immunofluorescent staining of endomucin demarcated the vasculature in both primary and secondary ossification centers and showed that Gli1[+] cells nestled among the metaphyseal blood vessels immediately under the growth plate but were clearly distinct from the endothelial cells (Fig. 4a, b). Immunostaining of intracellular aggrecan confirmed that the Gli1[+] cells were also separate from the growth plate chondrocytes expressing the proteoglycan (Fig. 4c, d). Although recent studies have shown that Lepr marks osteoblast progenitors in the bone marrow of adult mice, we found that few of the Gli1[+] cells beneath the growth plate (~7%) expressed Lepr detectable by immunostaining (Fig. 4e, f). As Pdgfra is a common marker for mesenchymal progenitors, we examined its potential expression in the Gli1[+] cells. For this, we took advantage of the Pdgfra-GFP allele that expressed GFP from the 3′-untranslated region of the endogenous Pdgfra locus and generated mice with the genotype of Gli1-CreER[T2]; Ai9; Pdgfra-GFP. Administration of TM at 1 month of age showed that most (~80%) of the Gli1[+] cells under the growth plate expressed Pdgfra and therefore were likely mesenchymal progenitors (Fig. 4g, h). To confirm the expression profile of Lepr and Pdgfra among the Gli1[+] cells, we isolated the cells from the chondro-osseous junction of the long bones at 24 h after TM and performed fluorescence-activated cell sorting (FACS) analyses with antibodies against the endogenous proteins. In keeping with the immunostaining results, among the tdTomato[+] mesenchymal population (CD31[−]CD45[−]Ter119[−]tdTomato[+]) <10% expressed Lepr but >60% was Pdgfra-positive (Fig. 4i). To gain further insight about the progenitors along the osteoblast lineage, we examined the expression of Runx2 and Osx and observed positive signal in ~70% of the Gli1[+] progenitors for Runx2 but only ~20% for Osx at 1 month of age (Fig. 4j–m). On the other hand, no Sox9 signal was detected in the Gli1[+] progenitors, indicating that they were separate from the chondrogenic lineage (Supplementary Figure 3A–C). In keeping with the progression of osteoblast differentiation, after 1 month of chasing,

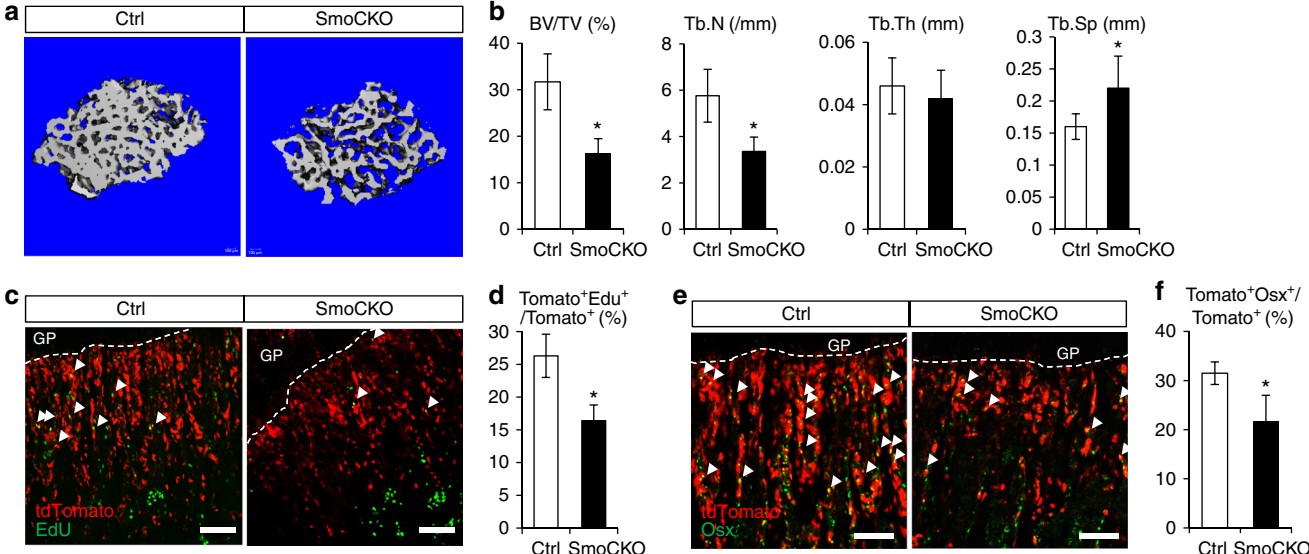

**Fig. 5** Hh signaling is required for normal MMP proliferation and differentiation. Gli1-CreER[T2];Ai9;Smo[c/c] (SmoCKO) or Gli1-CreER[T2];Ai9 (Ctrl) mice treated with TM at 1 month of age and harvested after 10 days. GP growth plate. Scale bar: 100 μm. **a, b** Representative images **a** or quantification of bone parameters **b** from μCT analyses of cancellous bone region of distal femur. **c** EdU staining on sections of proximal tibia. Arrowheads indicate EdU[+]tdTomato[+] cells. **d** Percentage (mean ± SD) of EdU[+]tdTomato[+] over tdTomato[+] cells quantified within cancellous bone region extending 300 μm from the growth plate of proximal tibia in two sections per mouse and three mice per genotype. **e** Immunofluorescence staining of Osx on sections of the proximal tibia. Arrowheads denote Osx[+]tdTomato[+] cells. **f** Percentage (mean ± SD) of Osx[+]tdTomato[+] over tdTomato[+] cells quantified within cancellous bone region extending 300 μm from the growth plate of proximal tibia in two sections per mouse and three mice per genotype. *$p < 0.05$, $n = 3$, paired Student's t-tests

the Osx[+] population among the Gli1-lineage cells increased to ~50% (Fig. 4n, o). The results so far therefore establish the Gli1[+] cells beneath the growth plate as osteogenic mesenchymal progenitors.

In our chasing experiments, we consistently observed a progressive increase of tdTomato[+] cells within the bone marrow proper. Whereas there was no tdTomato signal in the bone marrow of 1-month-old mice at 24 h after TM, positive cells were readily detectable after 9 months of chasing (Fig. 2j). We therefore decided to assess the potential expression of Lepr and Pdgfra among the tdTomato[+] BMSCs (CD31[−]CD45[−]Ter119[−] tdTomato[+]) by FACS analyses. In contrast to the Gli1[+] progenitors that expressed little Lepr, ~50% of the Gli1-lineage BMSCs were Lepr-positive after 6 months of chasing (Fig. 4p, left). On the other hand, the Pdgfra expression profile was similar between the Gli1-lineage bone marrow cells after chasing and the initial Gli1[+] progenitors, with both at ~60% (Fig. 4p, right, compare with Fig. 4i, right). Therefore, the Gli1[+] mesenchymal progenitors beneath the growth plate, besides being osteoblast precursors, are likely to produce BMSCs expressing Lepr or Pdgfra or both in adult mice. We next assessed whether the Gli1-lineage-derived BMSCs were enriched for fibroblast colony-forming units (CFU-F). For this, we performed CFU-F assays with BMSCs after 3 months of chasing following TM induction at 1 month of age. We found no difference between the percentage of tdTomato[+] CFU-F (2.1% ± 0.23%, SD, $n = 3$) and the percentage of tdTomato[+] BMSCs (1.8% ± 0.4%, SD, $n = 3$). Thus the Gli1-lineage-derived BMSCs exhibit a similar colony-forming ability to the bulk BMSCs.

As adipocytes are an important constituent of the adult bone marrow, we next investigated whether the Gli1[+] mesenchymal progenitors also produce bone marrow adipocytes in vivo. Perilipin immunofluorescent staining detected no signal in Gli1[+] cells in 1-month-old mice, indicating that mature adipocytes in the bone marrow did not experience Hh signaling (Fig. 4q, r). Remarkably, after 3 months of chasing, ~30% of the

perilipin[+] adipocytes located within the distal metaphyseal region of the femur expressed tdTomato, indicating their origin from the Gli1[+] cells (Fig. 4s, t). Taken together, the data so far demonstrate that the Gli1[+] mesenchymal progenitors can give rise to osteoblasts, BMSCs as well as bone marrow adipocytes and therefore are at least tri-potent in vivo.

The study so far has focused on 1 month of age, a stage when mice are rapidly growing. We next examined whether the Gli1[+] progenitor pool changes with aging. Because we noted that the cancellous bone mass notably decreased after 4 months of age, we administered TM to the Gli1-CreER[T2]; Ai9 mice at 1, 2, 4, or 12 months of age and monitored the Gli1[+] cells after 24 h. We performed immunofluorescent staining against aggrecan to mark the growth plate, a critical landmark for locating the Gli1[+] progenitors of interest. By using a modified protocol that detects both intracellular and extracellular aggrecan, we detected aggrecan deposition both within and below the growth plate in 1-month-old mice (Fig. 4u, v). This staining pattern was strikingly different from that of intracellular aggrecan shown earlier and marking only aggrecan-producing cells (Fig. 4c, d), thus supporting the view that cartilage matrix remnants serve as a scaffold for nascent bone matrix deposition during endochondral ossification. Importantly, at 1 month of age, numerous Gli1[+] cells were distributed among the aggrecan-containing matrix below the growth plate (Fig. 4v). By 4 months, however, few Gli1[+] cells were present under the growth plate, concurrent with the disappearance of the aggrecan-containing matrix from the cancellous bone region (Fig. 4w, x). At 12 months of age, essentially no Gli1[+] cells were detectable even though the growth plate was present (Fig. 4y, z). Thus the Gli1[+] mesenchymal progenitors beneath the growth plate are abundant in the young postnatal mice but diminish precipitously with age. To assess whether changes in proliferation could account for the decline in the Gli1[+] progenitor pool, we performed a proliferation assay by injecting 5-ethynyl-2′-deoxyuridine (EdU) into 1- or 2-month-old mice at 24 h after TM and 4 h before harvest. The results showed

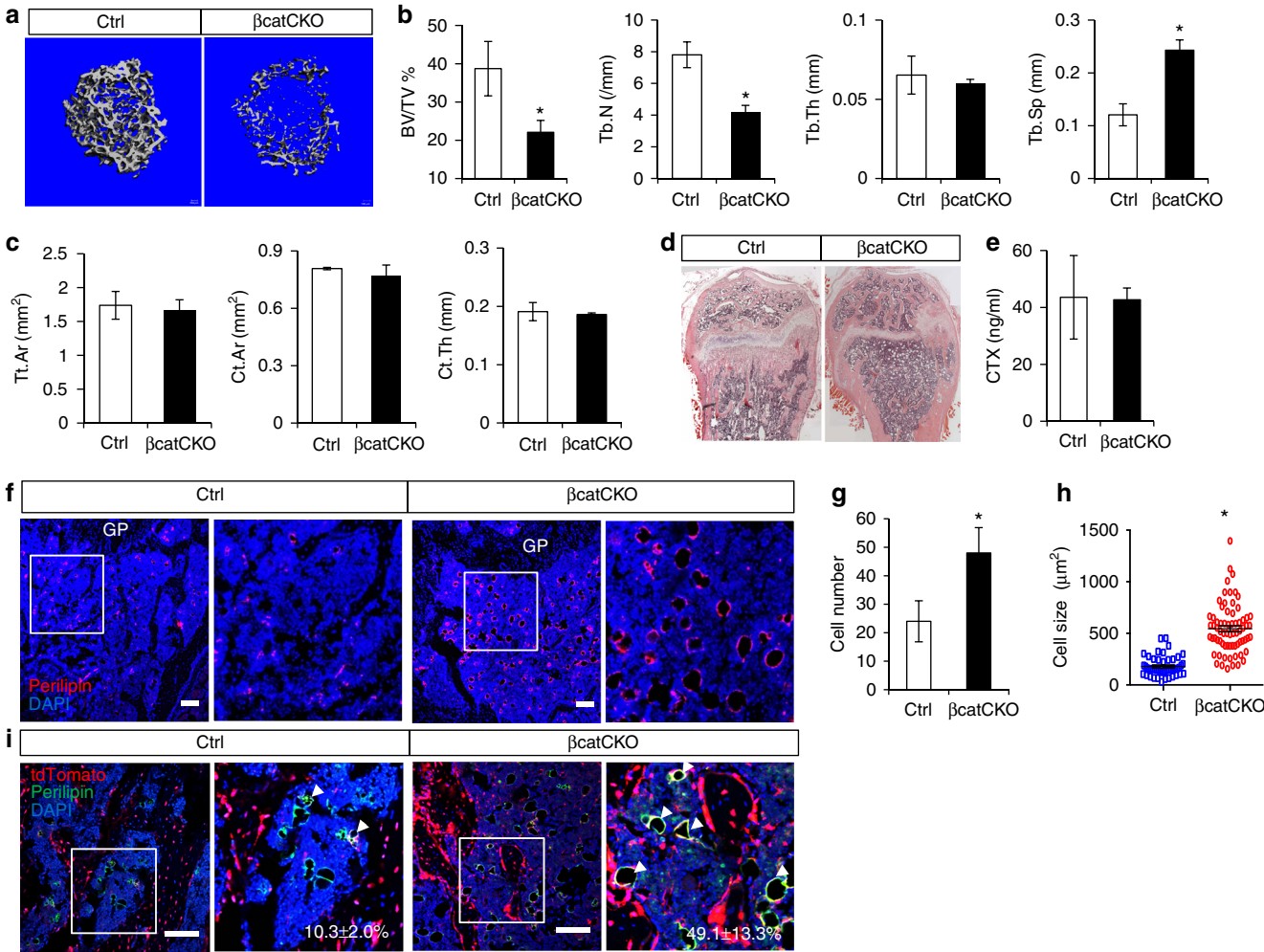

**Fig. 6** β-Catenin deletion in MMPs leads to osteopenia and fate switch toward adipogenesis. Gli1-CreER[T2]; Ai9 (Ctrl), or Gli1-CreER[T2];Ai9;β-catenin[c/c] (βcatCKO) littermate mice were treated with TM at 1 month of age and analyzed after 1 month. **a** μCT images of cancellous bone in distal femur. **b** Trabecular bone parameters from μCT analyses of distal femur. **c** Cortical bone parameters from μCT analysis of femoral diaphysis. Tt.A, total area, Ct.Ar cortical area, Ct.Th cortical thickness. **d** H&E staining for distal femur. **e** Serum CTX-I levels. **f** Immunofluorescence staining of Perilipin (red) on sections of distal femur. Boxed areas shown in magnified images to the right. Scale bar, 100 μm. GP growth plate. **g** Number of Perilipin[+] adipocytes within cancellous bone region extending 400 μm from the growth plate of distal femur in two sections per mouse and three mice per genotype. Data represent mean ± SD. **h** Area of Perilipin[+] adipocytes (meansured by imageJ) from the same area as **g**. Data was collected from two sections per mouse and three mice per genotype. A total of 80 adipocytes in each genotype (each represented by one data point) were quantified. **i** Confocal images showing tdTomato[+] cells (red) and immunofluorescence staining of Perilipin staining (green) on sections of distal femur. Boxed areas are shown in magnified images to the right. Scale bar: 100 μm. Arrowheads indicate Perilipin[+]tdTomato[+] adipocytes. Percentage (mean ± SD) of Perilipin[+]tdTomato[+] over Perilipin[+] adipocytes was calculated within cancellous bone region extending 400 μm from the growth plate in two sections per mouse and three mice per genotype. *$p < 0.05$, $n = 3$, paired Student's t-tests

that the Gli1[+] mesenchymal progenitors at the chondro-osseous junction exhibited a similar proliferation index of ~30% between 1 and 2 months of age, indicating that a mechanism other than proliferation is likely responsible for the rapid loss of the Gli1[+] progenitors with age (Supplementary Figure 2). We also detected a similar proliferation rate (~20%) among the Gli1[+] growth-plate chondrocytes between the two ages. In contrast, the Gli1[+] articular chondrocytes exhibited no proliferation (Supplementary Figure 2). As a previous study using the Nestin-GFP mice showed a similar age-dependent decrease for nestin[+] cells under the growth plate, we performed nestin immunostaining to examine a potential relationship between the Gli1[+] and the nestin[+] cells[28]. We, however, observed no overlap between the two populations in either 1-month- or 1-year-old mice (Supplementary Figure 3D, E). Overall, based on their anatomical location and their tri-

potential for differentiation in vivo, we designate the Gli1[+] cells immediately under the growth plate of juvenile mice MMPs.

To gain a better understanding of the molecular identity of MMPs, we compared its RNA profile with that of the Gli1[−] mesenchymal cells from the same region. Specifically, we performed RNA-seq experiments with FACS-sorted Gli1[+] (CD31[−]CD45[−]Ter119[−]tdTomato[+]) vs. Gli1[−] (CD31[−]CD45[−] Ter119[−]tdTomato[−]) cells from the chondro-osseous junction of the distal femur in 1-month-old Gli1-CreER[T2]; Ai9 mice harvested at 24 h after TM. The data revealed that the Gli1[+] MMPs were significantly enriched in mRNA for a number of markers previously assigned to MSCs, including CD44, CD106/ Vcam1, CD146/Mcam, Pdgfra, Pdgfrb, and αSma/Acta2 (Fig. 4aa, Table 1). Interestingly, Lepr mRNA was also enriched in MMPs even though immunostaining did not detect the protein in most

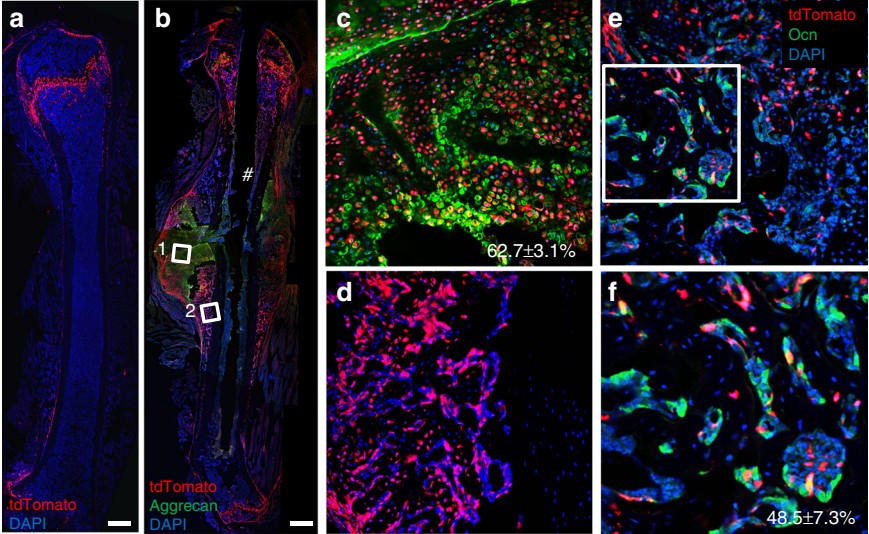

**Fig. 7** Gli1[+] cells produce chondrocytes and osteoblasts during fracture healing. Gli1-CreER[T2]; Ai9 mice were administered TM at 1 month of age and subjected to fractures 6 weeks later. Mice were harvested at 10 days after fracture. Scale bar: 500 μm. **a** Representative confocal image of contralateral (unfractured) femoral sections at the time of harvest. Note relatively little tdTomato signal (red) in periosteum. **b–d** Representative confocal image of femoral sections through fracture site, with immunofluorescence staining of Aggrecan (green) to demarcate chondrocytes. Hash symbol indicates position of pin used for fracture stabilization. Boxed areas of cartilage callus (box 1) and bony region (box 2) are shown at higher magnification in **c** and **d**, respectively. Percentage (mean ± SD, n = 3) of tdTomato[+]Aggrecan[+] over Aggrecan[+] cells in **c** was calculated from three mice (one representative section per mouse). **e**, **f** Immunofluorescence staining of Ocn on bony callus. Boxed area in **e** is shown at higher magnification in **f**. Percentage (mean ± SD, n = 3) of tdTomato[+];Ocn[+] over Ocn[+] osteoblasts was calculated from three mice (one representative section per mouse)

of the cells (Fig. 4aa, f). On the other hand, another MSC marker Sac-1/Ly6a, or the pericyte marker NG2/Cspg4, was not preferentially expressed by MMPs (Fig. 4aa, Table 1). Thus the Gli1[+] MMPs identified here express multiple but not all markers previously identified in mesenchymal stem/progenitor cells.

**Hh signaling controls MMP proliferation and differentiation.** Gli1 expression could simply be a marker for MMPs or could indicate a physiological function for Hh signaling in those cells. To distinguish those possibilities, we genetically deleted Smo, an obligatory mediator of Hh signaling, from the Gli1[+] cells in postnatal mice. Briefly, mice with the genotype of either Gli1-CreER[T2]; Ai9 (Ctrl) or Gli1-CreER[T2]; Ai9; Smo[c/c] (SmoCKO) were treated with TM once daily for 3 consecutive days starting at 1 month of age and harvested them at 10 days after the last TM dosing. The mice appeared to be normal and did not exhibit obvious distress at the time of harvest. Micro-computed tomography (μCT) imaging and quantification of the cancellous bone in the distal femur detected lower bone mass (BV/TV) and trabecular bone number (Tb. N) coupled with increased trabecular spacing (Tb. Sp) in SmoCKO over Ctrl mice (Fig. 5a, b). On the other hand, serum CTX-I levels were similar between the genotypes, indicating normal bone resorption in the SmoCKO mice. Thus Hh signaling is required for normal cancellous bone accrual in juvenile mice.

We next examined the cellular effect of Smo deletion in MMPs. The overall pool size of MMPs and their descendants (tdTomato[+]) was markedly reduced in the SmoCKO mice (Fig. 5c). EdU labeling assays revealed that Smo deletion decreased proliferation among the tdTomato[+] cells (Fig. 5c, d). In addition, Smo deletion diminished the percentage of the tdTomato[+] cells transitioned to the preosteoblast stage as marked by Osx expression (Fig. 5e, f). These results demonstrate that Hh signaling is necessary for normal proliferation and osteoblast differentiation of MMPs and/ or their progeny cells.

**β-Catenin deletion switches MMP fate towards adipogenesis.** We next examined the potential role of β-catenin in mediating osteoblast differentiation of MMPs. Previous work has established that Wnt/β-catenin signaling functions downstream of Hh to regulate osteoblastogenesis in the mouse embryo[29, 30]. In addition, postnatal deletion of β-catenin in Osx[+] cells not only reduced bone mass but also increased bone marrow adiposity[31, 32]. Here we treated mice with the genotype of either Gli1-CreER[T2]; Ai9 (Ctrl) or Gli1-CreER[T2]; Ai9; β-catenin[c/c] (βcatCKO) with TM once daily for 3 days starting at 1 month of age and analyzed them at 1 month after the last TM dosing. μCT imaging revealed that β-catenin deletion significantly reduced cancellous bone mass in the distal femur (Fig. 6a). Quantitative analyses indicated that the decrease in bone mass (BV/TV) was due to reduced trabecular number (Tb. N) but no change in trabecular thickness (Tb. Th) resulting in increased trabecular spacing (Tb. Sp) (Fig. 6b). On the contrary, the cortical bone parameters in βcatCKO mice were all normal (Fig. 6c). This result was consistent with the earlier observation that there was little targeting of the cortical bone by Gli1-CreER[T2] after 1 month of chasing (see Fig. 2f). Histology revealed a virtual lack of cancellous bone in the metaphysis of βcatCKO mice (Fig. 6d). The osteopenia was unlikely due to increased bone resorption as the serum CTX-I level was normal in the mutant mouse (Fig. 6e). These results therefore demonstrate that β-catenin in MMPs is critical for supporting cancellous bone formation in postnatal mice.

We next examined whether loss of β-catenin affected other differentiation fates of MMPs. Histology of the femur indicated a notable increase in adipocytes (identifiable by round and clear morphology) in place of cancellous bone at the metaphysis in βcatCKO mice (Fig. 6d). More importantly, the MMP-derived adipocytes as demarcated by tdTomato were both more numerous and bigger in the βcatCKO mouse over the control (Fig. 6f). Quantification confirmed that β-catenin significantly increased both number and size of the marrow adipocytes

(Fig. 6g, h). To assess specifically whether β-catenin deletion affected the relative contribution of MMPs to total marrow adipocytes, we performed immunofluorescence staining for perilipin to mark all mature adipocytes and quantified the percentage of tdTomato⁺perilipin⁺ cells (MMP-derived) over total perilipin⁺ cells. Whereas the MMPs normally produced ~10% of the marrow adipocytes after 1 month of chasing, they accounted for up to ~50% when β-catenin was deleted (Fig. 6i). Thus loss of β-catenin appears to alter the fate decisions of MMPs to favor adipogenesis at the expense of bone formation.

**Gli1⁺ cells contribute to fracture healing**. The data so far have demonstrated that Gli1 marks MMPs critical for normal bone formation in postnatal mice. To test whether Gli1⁺ cells in postnatal mice also contribute to bone regeneration, we created a semistabilized fracture at the diaphysis of the femur to observe fracture healing. We first marked the Gli1⁺ cells in Gli1-CreER^T2; Ai9 mice with TM at 1 month of age and then subjected the mice to fracture at 10 weeks of age, before harvesting them after 10 days of healing. In agreement with our earlier observation in non-injured mice, the contralateral non-fractured femur exhibited prominent tdTomato expression in the articular cartilage and the metaphysis but no signal in the cortical bone except for occasional areas of the periosteum (Fig. 7a). Following extraction, however, strong tdTomato was detected throughout the fracture callus including both the bony and cartilaginous regions (Fig. 7b–d). Immunofluorescent staining for intracellular aggrecan indicated that ~63% of aggrecan⁺ chondrocytes were also tdTomato⁺ (Fig. 7c). Similarly, immunostaining of osteocalcin (Ocn) showed that ~50% of the Ocn⁺ osteoblasts within a bony callus region expressed tdTomato (Fig. 7e, f). These data indicate that Gli1 marks a major skeletal progenitor pool contributing to both bone and cartilage formation during bone fracture healing in postnatal mice.

## Discussion

Through genetic lineage-tracing experiments, we have provided evidence that the Gli1⁺ cells are a predominant source for osteoblasts throughout the life of a mouse. Whereas the embryonic Gli1⁺ cells give rise to essentially all osteoblasts in both fetal and postnatal skeleton, a pool of postnatal Gli1⁺ cells immediately beneath the growth plate, termed MMPs in the present study, are essential for cancellous bone formation in juvenile mice. The study therefore identifies Gli1 as a common molecular marker among most if not all mesenchymal progenitors destined to become osteoblasts in the mouse.

Several features about MMPs are worth noting. The cells are at least tri-potent, producing osteoblasts, BMSCs as well as bone marrow adipocytes in vivo. By immunostaining, most MMPs express Pdgfra and Runx2 but not Lepr, nestin, Sox9, or Osx. RNA profiling reveals that MMPs express a number of MSC marker genes, including CD44, CD106/Vcam1, and CD146/Mcam. β-Catenin is a critical modulator for fate decisions in MMPs as deletion of β-catenin leads to excessive adipogenesis at the expense of osteoblast differentiation. MMPs are transient as they are abundant in juvenile mice but greatly diminished by 4 months of age when the cancellous bone is notably reduced. However, the number of MMP-derived marrow adipocytes and BMSCs increases with time, and about half of the MMP-derived BMSCs express Lepr detectable by immunostaining in the adult mice. Considering that Lepr⁺ BMSCs produce osteoblasts in older adult mice, it is reasonable to postulate that MMP-derived Lepr⁺ BMSCs may serve as long-term osteoblast progenitors. Overall, MMP cells on the one hand are transient progenitors to support cancellous bone formation in young mice but on the other hand

likely give rise to long-term bone marrow progenitors to produce bone marrow stroma, adipocytes as well as osteoblasts in adult mice.

Besides normal bone formation, postnatal Gli1⁺ cells also contribute to both chondrocyte and osteoblast production during bone fracture healing. It is difficult at present to discern the physical location of Gli1⁺ cells responsible for the de novo formation of chondrocytes and osteoblasts. Previous studies have indicated that both BMSCs and periosteal cells expressing alpha-smooth muscle actin (αSMA) contribute to the fracture callus[15]. Thus either MMP-derived BMSCs or the Gli1-lineage cells in the periosteum, or both, could produce osteoblasts and/or chondrocytes during fracture healing. Because we have observed only a small number of Gli1-lineage cells in the periosteum of the countralateral unfractured femur, we speculate that the Gli1-lineage periosteal cells might rapidly expand in response to the fracture. Finally, as we have observed Gli1-lineage cells in the adjacent skeletal muscle, it cannot be ruled out that skeletogenic progenitors could arrive from the muscle.

The current study has extended our understanding about Hh signaling in the skeleton. Although previous work has documented extensively the essential role of Hh signaling in osteoblast differentiation in the embryo, its physiological function in postnatal bone formation remains unclear. Here, by deleting Smo specifically in the Hh-responsive cells, we provide clear evidence that Hh signaling is necessary for modulating the MMP pool by promoting both proliferation and osteoblast differentiation in postnatal mice. Although we have not investigated the source of Hh protein in this setting, Ihh from the prehypertrophic chondrocytes is likely responsible for regulating the MMP population. Overall, our study supports the notion that all cells destined to become osteoblasts in the mouse likely experience Hh signaling at an early precursor stage. In this regard, it is interesting to note that Gli1-lineage cells were also a major source of osteoblast-like cells during vascular calcification in a mouse model of chronic kidney disease[33]. Although the molecular mechanism for this apparent "rite of passage" for osteoblasts remains to be elucidated in the future, the finding could be instructive for producing osteoblasts from naive mesenchymal progenitors in the context of regenerative medicine.

## Methods

**Mice**. Mouse strains Gli1-CreER^T2, Ai9, ColI-Gfp, Pdgfra-Gfp, Rosa-DTA, β-catenin^c/c, and Smo^c/c are as described[27, 34–39]. All mice were maintained in a specific pathogen-free facility, and Washington University Institutional Animal Care and Use Committee approved all experimental procedures.

**TM administration**. For embryonic studies, TM (Sigma) (4 mg/30 g body weight, dissolved in corn oil) was administered once by oral gavage to the dam at E13.5. For postnatal studies, TM (5 mg, dissolved in corn oil) was administered once daily for 3 consecutive days by oral gavage. Both males and females were used in lineage-tracing experiments and no sex-dependent difference was observed. Sex-matched littermates were used for comparison between experimental groups.

**Immunofluorescence staining**. Immunofluorescence staining was performed according to a published procedure[40]. Briefly, mice were perfused with 4% paraformaldehyde (PFA), and dissected bones were further fixed with 4% PFA overnight at room temperature and partially decalcified in 14% EDTA for 3 days with daily change of solution. The bones were then infiltrated with 30% sucrose overnight at 4 °C for cryoprotection and embedded in optimal cutting temperature (Tissue-Tek). Sections of 10-μm thickness were prepared with a Leica cryostat equipped with Cryojane (Leica, IL). Lepr staining was performed with sections of 80-μm thickness. The sections were kept at 20 °C until use. For immunostaining of both intracellular and extracellular aggrecan, slides were pretreated with hyaluronidase (Sigma, H4272, 2 mg/ml) at 37 °C for 20 min before antibody staining. The antibodies used in this study are as follows: Osx (ab22552, Abcam, 1:100), Endomucin (sc-65495, Santa Cruz Biotech. 1:100), Osteocalcin (sc-300045, Santa Cruz Biotech. 1:50), Aggrecan (ab1031, Millipore, 1:100), Perilipin (9349, Cell Signaling Technology, 1:100), and goat-anti-Lepr-biotin (R&D, 1:200). The secondary antibodies include goat anti-rabbit Alexa Fluor 647, goat anti-rabbit Alexa Fluor 488,

donkey anti-goat Alexa Fluor 488, and goat anti-rat Alexa Fluor 488 (Thermo-Fisher Scientific, 1:200). Slides were mounted with antifade mounting medium with DAPI (Vector Laboratories), and images were acquired with a confocal microscope (Nikon C-1 confocal system).

**Histology and histomorphometry.** For histology, femora and tibiae were isolated from mice after perfusion with 4% PFA and fixed in 10% buffered formalin overnight at room temperature, followed by decalcification in 14% EDTA for 2 weeks. After decalcification, femora and tibiae were processed for paraffin embedding and then sectioned at 6-µm thickness. Hematoxylin and eosin (H&E) staining was performed on paraffin sections according to the standard procedure.

**Preparation of cells for FACS analyses.** Whole bone marrow was gently flushed from the femur in a phosphate-buffered saline (PBS) buffer supplemented with 20% fetal bovine serum (FACS Buffer) and then treated with red cell lysis buffer at room temperature for 5 min. The bone marrow cells were then centrifuged and re-suspended in FACS Buffer. Cells from the cancellous bone region of long bones were isolated as follows. Upon removal of the epiphysis, the metaphyseal bone region extending about 5 mm from the growth plate was dissected and crushed with mortar and pestle in FACS Buffer, followed by digestion with 0.25% collagenase type I (C0130, Sigma-Aldrich) in PBS (Gibco) for 20 min at 37 °C with gentle shaking. The cells were then centrifuged and re-suspended with FACS Buffer and filtered with a 70-µm cell strainer.

**Flow cytometry.** Cells re-suspended in 100 µl FACS Buffer were incubated with Fc blocker anti-CD16/32 antibody (101319, Biolegend) for 10 min on ice, followed by staining with fluorochrome-conjugated antibody on ice for 30 min. The antibodies used in this study include anti-mouse CD140a (Pdgfra) APC (17-1401-81, eBioscience, 1:100), goat-anti-Lepr-biotin (AF497, R&D, 1:100), anti-Ter119-PE-cy7 (25-5921-81, eBioscience, 1:1000), anti-CD31-PE-cy7 (25-0311-81, eBioscience, 1:1000), and anti-CD31-PE-cy7 (25-0451-81, eBioscience, 1:1000). Samples stained with biotin-conjugated antibodies were washed with FACS Buffer and then incubated with streptavidin-brilliant violet 421 (Biolegend, 1:500) for 20 min. Flow cytometric analyses were carried out with an Lsr Fortessa flow cytometer equipped with the FACS Diva 6.1 software (BD Biosciences). Data were analyzed with FlowJo version 9.

**FACS sorting and RNA sequencing.** Cells from the cancellous bone region were isolated and stained as described above. The cells were then sorted with BD FACSAriaII-1 (Department of Pathology and Immunology, Washington University in St. Louis) for CD31⁻CD45⁻Ter119⁻tdTomato⁺ (Gli1⁺) cells vs. CD31⁻CD45⁻Ter119⁻tdTomato⁻ (Gli1⁻) cells. Sorted cells from multiple mice were pooled so that a sufficient amount of RNA can be extracted in each preparation with the NucleoSpin RNA XS Kit (740902.50, Macherey-Nagel). RNA from three paired pools of Gli1⁺ or Gli1⁻ cells (each pair from the same group of mice) was subjected to RNA-seq at the Genome Technology Access Center (GTAC, Washington University in St. Louis). RNA-seq reads were aligned to the Ensembl release 76 top-level assembly with STAR version 2.0.4b. Gene counts were derived from the number of uniquely aligned unambiguous reads by Subread:featureCount version 1.4.5. Transcript counts were produced by Sailfish version 0.6.3. Sequencing performance was assessed for total number of aligned reads, total number of uniquely aligned reads, genes and transcripts detected, ribosomal fraction known junction saturation, and read distribution over known gene models with RSeQC version 2.3. All gene-level and transcript counts were then imported into the R/Bioconductor package EdgeR and TMM normalization size factors were calculated to adjust for samples for differences in library size. Genes or transcripts not expressed in any sample were excluded from further analysis. The TMM size factors and the matrix of counts were then imported into R/Bioconductor package Limma and weighted likelihoods based on the observed mean–variance relationship of every gene/transcript and sample were then calculated for all samples with the voomWithQualityWeights function. Performance of the samples was assessed with a spearman correlation matrix and multi-dimensional scaling plots. Gene/transcript performance was assessed with plots of residual standard deviation of every gene to their average log-count with a robustly fitted trend line of the residuals. Generalized linear models were then created to test for gene/transcript level differential expression. Differentially expressed genes and transcripts were then filtered for false discovery rate-adjusted p-values ≤0.05. Heatmaps of select genes were generated from voom transformed values with Heatmap3 package from R/Bioconductor. The full dataset for RNA-seq was submitted to NCBI (SRA) (BioProjectID: PRJNA396564).

**MicroCT and in vivo microCT.** The femurs were scanned with a µCT system (µCT 40; Scanco Medical AG) following the recommendations by American Society for Bone and Mineral Research[41]. For quantifying trabecular bone parameters, 100 CT slices (1.6 mm total) immediately proximal to the distal femoral growth plate were analyzed. For quantifying cortical bone parameters, 50 slices (0.8 mm total) at the mid-diaphysis (starting from 6.8 mm proximal to the distal femoral articular surface) were analyzed. To monitor bone growth with time, in vivo microCT (vivaCT 40; Scanco Medical) was used to scan the femurs in live mice. For

quantifying trabecular bone parameters, 60 slices (1.2 mm total) immediately proximal to the the distal femoral growth plate were analyzed (threshold set at 220).

**Serum CTX-I and P1NP assays.** For serum cross-linked CTX-I and amino-terminal P1NP assays, serum was collected from mice after 6 h of fasting. Assays were performed with the RatLaps ELISA or Rat/Mouse P1NP EIA Kit (both from Immunodiagnostic Systems, Ltd.).

**Proliferation assay.** EdU was injected intraperitoneally at 10 µg/g body weight at 4 h before harvest. Frozen sections were subjected to a click reaction according to the manufacturer's instructions (Click-iT EdU Alexa Fluor 488 Imaging Kit, Invitrogen).

**Bone fracture.** Fracture-healing studies were performed with 10-week-old mice by using a semistabilized femur fracture model as previously described[42]. Briefly, anesthetized mice were prepared for surgery by shaving the fur around the knee area and disinfecting the skin with Betadine. An incision was made to dislocate the patella and the intramedullary canal was accessed by creating a hole in the distal end of the femur. A guidewire was placed into the femur and the mouse secured into custom three-point bending fixtures on a material testing machine (Dynamight 8841, Instron). The femur was broken using a controlled displacement ramp. Then the femur was stabilized with a 24-gauge stainless steel pin that was positioned over the guidewire. The guidewire was removed, rod trimmed, and the wound sutured and closed. Animals were given Buprenex to alleviate any surgical pain. Radiographs were taken immediately following surgery and prior to sacrifice to confirm pin stability. Mice were analyzed 10 days after bone fracture.

**Statistics.** Quantitative data were evaluated by paired Student's t-tests. p < 0.05 was considered statistically significant.

**Data availability.** The full dataset for RNA-seq supporting the molecular profile of Gli1⁺ MMP is available at NCBI (SRA) (BioProjectID: PRJNA396564).

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

## Acknowledgements

The work is supported in part by NIH grants AR060456 (to F.L.), AR055923 (to F.L.), AR89290 (to M.J.S.), and AR89215 (to M.J.S.). We also acknowledge support of Washington University Center for Musculoskeletal Research funded by NIH P30 AR057235. We thank Eric Tycksen (GTAC, Washington University School of Medicine) for his generous help with RNA-seq data analyses.

## Author contributions

Y.S., G.H., W.-C.L., and J.A.M. conducted experiments. M.J.S. directed fracture procedure. F.L. oversaw the project and wrote the paper.

## Additional information

**Competing interests:** The authors declare no competing financial interests.

