## [Peer Review File · Nature Communications]

Reviewers' comments:

Reviewer #1 (Remarks to the Author):

Previous studies demonstrated that Gli1 marks “mesenchymal stem cells” in the craniofacial bones, the incisor as well as several internal organs of adult mice. However, whether Gli1 marks skeletal stem/progenitor cells in the long bones remains unexplored. In this manuscript by Shi Y et al., the authors performed a very detailed characterization on the localization and lineage fate of Gli1+ cells in fetal and postnatal bone mainly using lineage-tracing experiments. The authors found that the Gli1+ cells give rise to essentially all osteoblasts in both fetal and postnatal skeleton. Specifically, a pool of postnatal Gli1+ cells immediately beneath the growth plate, are essential for cancellous bone formation in young adult mice. In addition, postnatal Gli1+ cells contribute to chondrocytes and osteoblasts during bone fracture healing. This is a significance study as it demonstrates that Gli1 can be used as a common molecular marker to label most mesenchymal progenitors destined to become osteoblasts in fetal and postnatal long bone. The authors have provided abundant evidences to support their conclusions. The authors may consider the following minor comments.

1. It is unclear from the evidence provided whether the properties of the Gli1+ cells in different location of the fetal and postnatal long bone are different. For instance, are the Gli1+ cells immediately beneath the growth plate relative to those at other sites have higher CFU-F activity? Are the Gli1+ Lepr+ cell population relative to Gli1+ Lepr- cell population have higher CFU-F activity? In situ analysis of BrdU labeling would also help to evaluate the proliferative /quiescent property of the Gli1+ cells at different locations and at different developmental stages in postnatal long bone.

2. The finding that the Gli1+ cells beneath the growth plate are abundant in the young postnatal mice but diminish precipitously with age is very interesting. It was previously reported that Nestin-expressing and Nestin-GFP+ cells at the same region (primary spongiosa) of long bones underwent quite similar age-dependent change (Dev Cell 2014; 29:330-339). It would be interesting to assess whether or not Gli1 and Nestin are co-expressed in most of the cells immediately beneath the growth plate in postnatal long bone and diminish with age.

Reviewer #2 (Remarks to the Author):

This study by Shi et al., using a genetic lineage-tracing experimental approach shows that Gli1+ positive cells are the main source of osteoblasts through the entire life-span of mouse. The authors identified a Gli1+ population of progenitors cells which differentiate into osteoblasts participating to both, fetal and postnatal skeleton scaffold. Furthermore, they provide evidences that a pool of Gli+ cells resident immediately below the growth place, referred to as metaphyseal mesenchymal progenitors (MMPs) play a crucial role for

cancellous bone formation in adult mouse.

The work is overall well performed and results convincing. Outcomes highlighting a Gli1+ cell population as progenitors rather than "mesenchymal stem cells" provide a rigorous and better biological function and definition of Gli1+ positive cell than that previously reported by Chai group (Nat. Cell Biol., 2015)

However, the reviewer has some questions and suggestions for the authors:

1. At least for Figure 1, the authors should include a pentachrome-Movat's staining of sister slides to allow a better appreciation of each anatomical part of their specimens (e.g. growth plate, epiphysis and so on). Since, they corroborate the initial finding also later in other figures it is important to present at beginning a better anatomical representation of bones.
2. Figure 2H, shows a cranial suture. Could the authors specify which suture precisely is that? Based on what it stated in the figure legend 2: "Representative confocal image showing a sagittal section of skull" " the review understands that is a coronal (COR) suture section specimen. Moreover, the authors showed only one time point (9 months after induction). What is the fate/location of the Gli1+ cells within the suture at different time (earlier) points? And what about Gli1+ cells in other cranial sutures? The authors should investigate and follow up these aspects. A deeper description/analysis of Gli1+ cells resident in the cranial suture and vertebra as well is needed. Another important aspect to investigate: what the status of Gli1+ cells in a fusing suture such as the Posterior Frontal (PF) suture? Since the authors show that Gli1+ cell produce bot chondrocytes and osteoblasts and because the PF suture undergoes to fusion through endochondral ossification by day pN 15 in mouse, this specific suture, therefore, represents an ideal skeletal environment for this study.
3. A previous paper by Zhao et al., (Nature Cell Biology, 2015) has identify Gli1+ cell within the Sagittal (SAG) cranial suture and established this population as "stem cells" . How the authors correlate their Gli1+ cells to this previously described population? In few words, are they dealing with the same population or their cells represent a distinct population? Have they isolated the cells and tested them for self-renewal, CFU and lineage differentiation ability? Because in this study the authors provide also evidence of Gli1+ cells resident in the Sagittal it would be of interest to understand is these cells are or not the some cell population reported in the earlier study by Zhao et al. Can the authors exclude that are stem/progenitors cells? The above suggested experiment will answer the question.
4. Although the in vivo analysis of the current study is straightforward and provides evidences for their role in skeleton formation, repair and homeostasis, it must be pointed out that Gli1+ cells have already been identified as cells playing a role in osteogenesis and bone repair, as well as the crucial role of Hh-signaling in modulating osteoprogenitors/stem cells function in bone repair (Tevlin R et al., Sci. Transl. Med., 2017). Therefore, in order to add novelty and applicative potential to the current study, it would be worth and desirable to characterize at molecular and transcriptomic level the identified Gli1+ population. This could be, for example, easily achieved by FACS isolation of Gli1+ cells present beneath the growth plate. By gaining molecular and transcriptomic profiles of the Gli1+ cells will indeed, strength the findings reported herein and furthermore, will help to translate the findings into regenerative medicine application through the knowledge of novel and unveiled

mechanisms.

Minor points:

Please use the word juvenile instead of "young adult" mice.

Reviewer #3 (Remarks to the Author):

Concerns related to the paper:

1. The Authors have obtained an elegant series of Gli cells images at different ages. However, the conclusion that these cells, designated metaphyseal mesenchymal progenitors, are a source of "progenitors" in both early developmental and adult mice is an over statement. Further characterization is required to define how these cells are distinct at molecular genomic levels when compared to a pluripotent mesenchymal stem cell in the bone marrow or to a mesenchymal stem cell in the periosteum or in the perichondrium. The studies do not provide a true lineage characterization in terms of hierarchy for these mesenchymal stem cells. The major contribution of the paper is the exquisite imaging of the cells at different ages, but nearly all figures provide descriptive parameters of cell locations and bone loss. Although the authors show, for the first time, effects in the adult skeleton, the impact of the present paper is incremental to the vast knowledge of Gli1-signaling that is critical for bone formation. The novel contribution of the study is the potential for identifying the hierarchical lineage of the differently marked Gli1+ cells within the context of various tissues that comprise the long bone.
2. If these cells are a unique population, clarification is necessary to define distinct functional roles for these cells as early chondrocyte progenitors in the articular zone compared to cells in the calcified zone primary spongiosa. There is no molecular data to distinguish the properties of Gli1 positive MMPs from the endogenous BMsC and Gli+ cells that differentiate to chondrocytes vs Perichondrium MSCs.
3. The descriptions of cell surface markers of MSCs should be expanded. It is surprising that well-known markers (eg. SMA or several CD's) that identify the "classic MSCs" were not immunohistochemically stained or FACS-identified.
4. There is a striking absence in the manuscript of the transcriptional contribution of RUNX2 to these cells. There are several published papers that report the Gli regulation of RUNX and the RUNX regulation of Gli. Could the chondrocyte-derived cells from the population of Gli+ cells express none or low RUNX whereas in MMPs high Runx2 has been reported. It has been published that RUNX2 and RUNX3 play key roles in the differentiation to osteoblasts in the primary spongiosa. It is surprising that there are no studies in the paper that actually isolate the cells to compare phenotypic differences for some well-established markers that discriminate between the MMPs and BMSCs where the major effect occurs.

5. Three experiments were performed to support the conclusion that Gli1+ cells differentiate into osteoblasts. However, there are no mechanistic studies.

Reviewer #1:

This is a significance study as it demonstrates that Gli1 can be used as a common molecular marker to label most mesenchymal progenitors destined to become osteoblasts in fetal and postnatal long bone. The authors have provided abundant evidences to support their conclusions.

We thank the reviewer for positive comments.

1. It is unclear from the evidence provided whether the properties of the Gli1+ cells in different location of the fetal and postnatal long bone are different. For instance, are the Gli1+ cells immediately beneath the growth plate relative to those at other sites have higher CFU-F activity? Are the Gli1+ Lepr+ cell population relative to Gli1+ Lepr- cell population have higher CFU-F activity? In situ analysis of BrdU labeling would also help to evaluate the proliferative /quiescent property of the Gli1+ cells at different locations and at different developmental stages in postnatal long bone.

Postnatal labeling of the long bone has identified four anatomically distinct Gli1+ populations, namely the superficial zone cells of the joint, chondrocytes within the growth plate, perichondrial cells next to the prehypertrophic cartilage, and cells immediately beneath the growth plate. The present study focuses only on the last Gli1+ population (termed metaphyseal mesenchymal progenitors or “MMP” here) as it contains the progenitors for the trabecular bone osteoblasts and also bone marrow stromal cells and adipocytes. According to the reviewer’s recommendation, we have attempted to compare the CFU-F activity between the Gli1-lineage Lepr+ versus Lepr- cells, but have failed to culture the cells after FACS sorting. Therefore we have resorted to an alternative approach. We took an aliquot of the total bone marrow stromal cells (BMSC) to measure the percentage of Gli1-lineage cells (tdTomato+) by FACS analyses, and then used the rest of BMSC for CFU-F assays and determined the percentage of tdTomato+ colonies by fluorescence microscopy. We found that the two percentages were very similar (1.8%±0.4% and 2.1%±0.23%, respectively), and therefore conclude that the Gli1-lineage BMSC has a similar CFU-F activity as the rest of BMSC (page 12).

Based on the reviewer’s recommendation, we have also performed EdU labeling experiments following tamoxifen induction at either one or two months of age. These experiments revealed a similar EdU labeling index (~20-30%) between the Gli1+ growth plate chondrocytes and the Gli1+ cells immediately beneath the growth plate, but detected no labeling among the Gli1+ superficial layer cells at the joint (Fig. S2, supplemental figure). Furthermore, we saw no difference in the labeling index between the ages.

2. The finding that the Gli1+ cells beneath the growth plate are abundant in the young postnatal mice but diminish precipitously with age is very interesting. It was previously reported that Nestin-expressing and Nestin-GFP+ cells at the same region (primary spongiosa) of long bones underwent quite similar age-dependent change (Dev Cell 2014; 29:330-339). It would be interesting to assess whether or not Gli1 and Nestin are co-expressed in most of the cells immediately beneath the growth plate in postnatal long bone and diminish with age.

We thank the review for the insightful comment. We have now performed Nestin

immunofluorescence staining on bone sections from Gli1-CreER^{T2};Ai9 mice (one month or one year old) harvested at 24 hrs after tamoxifen injection, but detected little overlap between Nestin and Gli1 expression (Fig. S3B, C). Note that in one-year-old mice, few if any Gli1+ cells were present immediately beneath the growth plate although Nestin+ cells were still present.

Reviewer #2:

The work is overall well performed and results convincing. Outcomes highlighting a Gli1+ cell population as progenitors rather than “mesenchymal stem cells” provide a rigorous and better biological function and definition of Gli1+ positive cell than that previously reported by Chai group (Nat. Cell Biol., 2015)

We appreciate the reviewer’s encouraging comments.

1. At least for Figure 1, the authors should include a pentachrome-Movat’s staining of sister slides to allow a better appreciation of each anatomical part of their specimens (e.g. growth plate, epiphysis and so on). Since, they corroborate the initial finding also later in other figures it is important to present at beginning a better anatomical representation of bones.

We thank for review for the suggestion. We have now included H&E staining for a better appreciation of the anatomy (Fig. 1A’-C’, E’).

2. Figure 2H, shows a cranial suture. Could the authors specify which suture precisely is that? Based on what it stated in the figure legend 2: “Representative confocal image showing a sagittal section of skull” the review understands that is a coronal (COR) suture section specimen. Moreover, the authors showed only one-time point (9 months after induction). What is the fate/location of the Gli1+ cells within the suture at different time (earlier) points? And what about Gli1+ cells in other cranial sutures? The authors should investigate and follow up these aspects. A deeper description/analysis of Gli1+ cells resident in the cranial suture and vertebra as well is needed. Another important aspect to investigate: what the status of Gli1+ cells in a fusing suture such as the Posterior Frontal (PF) suture? Since the authors show that Gli1+ cell produce bot chondrocytes and osteoblasts and because the PF suture undergoes to fusion through endochondral ossification by day pN15 in mouse, this specific suture, therefore, represents an ideal skeletal environment for this study.

We appreciate the reviewer’s expertise. The original skull section (now Figure 2I) shows the lambdoid suture after 9 months of induction. The main point there is to demonstrate that the Gli1-lineage cells contribute to osteocytes in the parietal and interparietal bones after the extended time. As per the reviewer’s suggestion, we now also include data from the same suture harvested at 24 hrs after induction, which shows the presence of Gli1+ cells mainly within the interior (central layers) of the suture (new Fig. 2H). Examination of the PF suture at 24 hrs after induction showed similar distribution of Gli1+ cells within the suture (Fig. S1). A deeper analysis of the Gli1+ cells in the cranial sutures was not conducted here as they were previously studied in detail by the Chai group (Nat. Cell Biol., 2015).

3. A previous paper by Zhao et al., (Nature Cell Biology, 2015) has identify Gli1+ cell within the Sagittal (SAG) cranial suture and established this population as “stem cells”. How the authors correlate their Gli1+ cells to this previously described population? In few words, are they dealing with the same population or their cells represent a distinct population? Have they isolated the cells and tested them for self-renewal, CFU and lineage differentiation ability? Because in this study the authors provide also evidence of Gli1+ cells resident in the Sagittal it

would be of interest to understand is these cells are or not some cell population reported in the earlier study by Zhao et al. Can the authors exclude that are stem/progenitors cells? The above suggested experiment will answer the question.

Both studies used the same lineage-tracing tool Gli1-CreER^{T2} to identify the Gli1+ cells. Therefore the Gli1+ cells within the cranial sutures between the two studies are the same population. However, the reviewer raises an interesting question whether the Gli1+ mesenchymal progenitors in the long bones (the focus of the present study) are similar to those in the cranial sutures. We have shown that the long bone Gli1+ progenitors have CFU ability in vitro and differentiate into osteoblasts, adipocytes and bone marrow stromal cells in vivo, but did not compare them with the cranial Gli1+ cells in side-by-side in vitro assays. We believe that a comparison of transcriptome between the two populations by RNA-seq in the future will likely provide insights about their potential similarities at the molecular level.

4. Although the in vivo analysis of the current study is straightforward and provides evidences for their role in skeleton formation, repair and homeostasis, it must be pointed out that Gli1+ cells have already been identified as cells playing a role in osteogenesis and bone repair, as well as the crucial role of Hh-signaling in modulating osteoprogenitors/stem cells function in bone repair (Tevlin R et al., Sci. Transl. Med., 2017). Therefore, in order to add novelty and applicative potential to the current study, it would be worth and desirable to characterize at molecular and transcriptomic level the identified Gli1+ population. This could be, for example, easily achieved by FACS isolation of Gli1+ cells present beneath the growth plate. By gaining molecular and transcriptomic profiles of the Gli1+ cells will indeed, strength the findings reported herein and furthermore, will help to translate the findings into regenerative medicine application through the knowledge of novel and unveiled mechanisms.

We thank the reviewer for the constructive suggestion. We have now performed RNA-seq for the Gli1+ CD31-CD45-Ter119- (Gli1+) versus Gli1- CD31-CD45-Ter119- (Gli1-) cells that were FACS sorted from the chondro-osseous junction beneath the growth plate. The results reveal that the Gli1+ cells express multiple but not all MSC markers as previously described (Fig. 4O).

Minor points:

Please use the word juvenile instead of “young adult” mice.

Done.

Reviewer #3 (Remarks to the Author):

Concerns related to the paper:

1. The Authors have obtained an elegant series of Gli cells images at different ages. However, the conclusion that these cells, designated metaphyseal mesenchymal progenitors, are a source of “progenitors” in both early developmental and adult mice is an over statement. Further characterization is required to define how these cells are distinct at molecular genomic levels when compared to a pluripotent mesenchymal stem cell in the bone marrow or to a mesenchymal stem cell in the periosteum or in the perichondrium. The studies do not provide a true lineage characterization in terms of hierarchy for these mesenchymal stem cells. The major contribution of the paper is the exquisite imaging of the cells at different ages, but nearly all figures provide descriptive parameters of cell locations and bone loss. Although the authors show, for the first time, effects in the adult skeleton, the impact of the present paper is incremental to the vast knowledge of Gli1-signaling that is critical for bone formation. The novel contribution of the study is the potential for identifying the hierarchical lineage of the differently marked Gli1+ cells within the context of various tissues that comprise the long bone.

Our conclusion about the Gli1+ cells being a source of “progenitors” in both early developmental and adult mice is based on the following functional evidence in vivo: 1) when the Gli1+ cells were marked in the embryo (E13.5), they can be traced to all osteoblasts and osteocytes throughout the skeleton (long bones, spine, skull bones) in 2-month-old mice (Fig. 1); 2) when the Gli1+ cells were marked in juvenile mice (one month of age), the Gli1+ cells immediately beneath the growth plate (please note that only those Gli1+ cells are termed metaphyseal mesenchymal progenitors) give rise to essentially all of the trabecular bone osteoblasts that subsequently form, as well as some bone marrow adipocytes and stromal cells (Fig. 3, 4).

The reviewer raises an important point about the relationship between the Gli1+ cells and the bone marrow or perichondrial/periosteal MSC as previously described. Considering that the Gli1+ cells marked at E13.5 are lineage-traced to all osteoblasts and osteocytes in the cortical bone of adult mice, we believe that the perichondrial/periosteal MSC responsible for normal appositional growth of long bones are descendants of embryonic Gli1+ perichondrial cells (Fig. 1A). In addition, we show that the postnatal Gli1+ cells immediately below the growth plate (metaphyseal mesenchymal progenitors) can give rise to the Lepr+ BMSC, the main contributor to adult bone formation in vivo as previously reported.

To further address the reviewer’s concern, we have performed additional experiments to better define the molecular signature of the postnatal Gli1+ metaphyseal mesenchymal progenitors (MMP). Specifically, we performed RNA-seq experiments with the Gli1+ MMP in comparison with the Gli1- mesenchymal cells isolated from the same tissue area (immediately beneath the growth plate) of one-month-old mice. Those analyses show that the Gli1+ MMP are enriched with several known MSC markers, including CD44, Vcam1/CD106, Mcam/CD146, Pdgfra, Pdgfrb and Lepr but not some others (e.g., CD105, Sca-1/Ly6a, nestin, Cspg4/Ng2) (Fig. 4O).

2. If these cells are a unique population, clarification is necessary to define distinct functional roles for these cells as early chondrocyte progenitors in the articular zone compared to cells in the calcified zone primary spongiosa. There is no molecular data to distinguish the properties of

Gli1 positive MMPs from the endogenous BMSC and Gli+ cells that differentiate to chondrocytes vs Perichondrium MSCs.

The Gli1+ cells in the articular zone and the Gli1+ chondrocytes within the growth plate are distinct populations from the Gli1+ metaphyseal mesenchymal progenitors (MMP). Whereas the first two populations represent differentiated cell types (articular or growth plate chondrocytes) responsive to Hh signaling, the MMP are unique in that they are the progenitors for trabecular osteoblasts as well as bone marrow stromal cells and adipocytes. We do not have evidence that Gli1+ cells differentiate to chondrocytes. We have now provided RNA-seq data that the postnatal Gli1+ MMP is enriched with a number of markers previously assigned to MSC (Fig. 4O). The embryonic Gli1+ cells labeled at E13.5 are precursors for all osteoblasts/osteocytes and the perichondrial/periosteal cells (presumably including perichondrium MSCs as the reviewers refer to), but they are not the focus here and thus not characterized molecularly.

3. The descriptions of cell surface markers of MSCs should be expanded. It is surprising that well-known markers (eg. SMA or several CD's) that identify the "classic MSCs" were not immunohistochemically stained or FACS-identified.

RNA-seq now shows that the MMP are enriched with mRNA transcripts for CD146/Mcam, CD44, CD106/Vcam1 and α SMA/Acta2 (Fig. 4O).

4. There is a striking absence in the manuscript of the transcriptional contribution of RUNX2 to these cells. There are several published papers that report the Gli regulation of RUNX and the RUNX regulation of Gli. Could the chondrocyte-derived cells from the population of Gli+ cells express none or low RUNX whereas in MMPs high Runx2 has been reported. It has been published that RUNX2 and RUNX3 play key roles in the differentiation to osteoblasts in the primary spongiosa. It is surprising that there are no studies in the paper that actually isolate the cells to compare phenotypic differences for some well-established markers that discriminate between the MMPs and BMSCs where the major effect occurs.

Thanks to the reviewer's suggestion, we have now performed immunostaining for Runx2. The results show that ~73% of the Gli1+ MMP in the primary spongiosa co-express Runx2 (Fig. 4F1). However, Runx2 signal was not confined to the Gli1+ MMP in the primary spongiosa, indicating that Gli is unlikely to be the sole factor for inducing or maintaining Runx2 expression.

We currently do not have evidence that the Gli1+ chondrocytes are derived from a Gli1+ progenitor. The Gli1+ chondrocytes are accounted for by active Hh signaling in those cells at the time of tamoxifen induction, as it is well established that the proliferative chondrocytes in the growth plate respond to Ihh produced by the prehypertrophic and early hypertrophic chondrocytes.

As per the reviewer's suggestion, we have now isolated the postnatal Gli1+ MMP for RNA-seq and shown the enrichment of several MSC markers (Fig. 4O).

5. Three experiments were performed to support the conclusion that Gli1+ cells differentiate into osteoblasts. However, there are no mechanistic studies.

Although a mechanistic understanding about osteoblast differentiation from the Gli1+ MMP in vivo is clearly necessary, we believe that such studies are beyond the scope of the current study.

REVIEWERS' COMMENTS:

Reviewer #1 (Remarks to the Author):

My concerns have been well addressed.

Reviewer #2 (Remarks to the Author):

The authors have addressed properly point-by-point the reviewer's questions
And added new data as requested.

Reviewer #3 (Remarks to the Author):

Accept